# Photoelectrocatalytic Degradation of Methylene Blue on Electrodeposited Bismuth Ferrite Perovskite Films

**DOI:** 10.3390/ma16072769

**Published:** 2023-03-30

**Authors:** Oluchi V. Nkwachukwu, Charles Muzenda, Kehinde D. Jayeola, Tsholofelo I. Sebokolodi, Dimpo S. Sipuka, Marc Cretin, Minghua Zhou, Duduzile Nkosi, Omotayo A. Arotiba

**Affiliations:** 1Department of Chemical Sciences, University of Johannesburg, Doornfontein 2028, South Africa; 2Centre for Nanomaterials Science Research, University of Johannesburg, Doornfontein 2028, South Africa; 3IEM (Institute Européen des Membranes), UMR 5635, CNRS, ENSCM, UM, Université de Montpellier, Place E. Bataillon, F-34095 Montpellier CEDEX 5, France; 4Tianjin Key Laboratory of Environmental Technology for Complex Trans-Media Pollution, College of Environmental Science and Engineering, Nankai University, Tianjin 300350, China

**Keywords:** bismuth ferrite perovskite, electrodeposition, photoelectrocatalytic degradation, methylene blue, citric acid, film thickness

## Abstract

Electrodeposited bismuth ferrite (BiFeO_3_) thin films on fluorine-doped tin oxide (FTO) substrate were employed as photoanodes in the photoelectrocatalytic degradation of methylene blue. The BiFeO_3_ thin films electrodeposited for 300 s, 600 s, 1200 s, 1800 s and 3600 s were characterised with XRD, field emission scanning electron microscopy (FESEM) and UV–vis diffuse reflectance spectroscopy. SEM images displayed different morphology at different electrodeposition times which affects the photoelectrocatalytic (PEC) performances. The FESEM cross-sectional area was used to measure the thickness of the film. The optical properties showed that the band gaps of the photoanodes were increasing as the electrodeposition time increased. The photocurrent response obtained showed that all thin film photoanodes responded to visible light and lower charge transfer resistance (from electrochemical impedance spectroscopy studies) was observed with photoanodes electrodeposited at a shorter time compared to those at a longer time. The PEC application of the photoanode for the removal of methylene blue (MB) dye in water showed that the percentage degradation decreased with an increase in electrodeposition time with removal rates of 97.6% and 70% observed in 300 s and 3600 s electrodeposition time, respectively. The extent of mineralisation was measured by total organic carbon and reusability studies were carried out. Control experiments such as adsorption, photolysis, photocatalysis and electrocatalysis processes were also investigated in comparison with PEC. The electrodeposition approach with citric acid exhibited improved electrode stability while mitigating the problem of catalyst leaching or peeling off during the PEC process.

## 1. Introduction

Over the past few decades, the level of environmental pollution has increased tremendously and consequently, the treatment of industrial wastewater has received much attention. Surface and groundwater are more prone to pollution (especially from organic substances) owing to the inflow of poorly treated textile, paper and pharmaceutical industrial effluents from wastewater treatment plants [1,2]. Organic dyes such as methylene blue are major pollutants in the environment owing to their wide application in the textile and pharmaceutical industries [1]. It is necessary to develop methods to treat or minimise the concentration of organic pollutants in wastewater before discharge into receiving waters to reduce their harmful effect on humans and the environment. So far, advanced oxidation methods (which include electrochemical advanced processes) have shown promising results as alternative or complementary ways of removing organic pollutants from water or wastewater. The emergence of interest in visible light active materials owing to the drawbacks of some semiconductors such as TiO_2_ has geared research efforts towards perovskites. Perovskite is a material with a chemical structure like CaTiO_3_. Some perovskite and perovskite-like structures exhibit properties such as excellent dielectric, optical, ferroelectricity, superconductivity, piezoelectricity, multiferroicity and good magneto resistance properties [3,4,5]. Among the perovskites, bismuth ferrite (BiFeO_3_, BFO) is a multiferroic material with the ability to exhibit ferroelectric, ferromagnetic and ferroelastic ordering concurrently. It is a visible light active material with a narrow band gap (2.2–2.8 eV) [6,7]. It possesses a curie and Néel temperature at around 770 °C and 370 °C, respectively. BiFeO_3_ (BFO) forms a rhombohedral (R3c) structure in its crystallised state [8]. Owing to these properties BFO has found wide applications in water splitting [9,10], storage memory devices [11], water remediation [12,13] and solar cells [14,15]. Furthermore, BFO is tuneable with other transition metals and lanthanides because of the inter-linkage between its magnetism, charge and structural properties [16] bringing forth more improved and enhanced properties [17]. Numerous studies have reported the different ways to synthesise BiFeO_3_ nanostructure for various applications. Sati et al. [16] synthesised Pr and Ti codoped BFO multiferroic using a conventional solid-state method to investigate its properties. They observed improvement in dielectric and magnetic properties. Jamshaid et al. [18] used a solvothermal method to prepare a bentonite-Co-doped BFO for the photocatalytic degradation of methyl orange in water. Several methods such as pulsed laser [19], molecular beam epitaxy [20], sputtering [21], co-precipitation [22] and chemical bath [8,23] deposition techniques have been employed to grow BiFeO_3_ powder/films.

Photoelectrocatalytic (PEC) degradation is an important electrochemical wastewater treatment technology that involves the use of various materials, principally semiconductors, in the preparation of photoanodes [24,25,26,27]. The stability of the photoanode is an important factor in the photoelectrocatalytic treatment of organic pollutants in wastewater. Thus, novel synthetic routes that will reduce catalyst leaching [28,29,30,31] and improve catalyst activity on the anodic substrate when exposed to voltage and light are sought. Towards the preparation of BFO-based photoanode, the electrodeposition method from a citric acid chemical bath is reported in this study. Chemical deposition and electrodeposition techniques have attracted considerable attention due to their cost effectiveness, energy efficiency, environment compatibility [32], and facile and homogeneity of deposited material with large area coverage [33]. This method enables the deposition of films directly onto a substrate with a simple setup. The film thickness can be quantified over deposition time or by viewing the cross-sectional area of the films over a microscope. Studies of BiFeO_3_ synthesised by electrochemical deposition technique have not been extensively explored [8,23]. 

Citric acid (CA) is a chelating agent often used to buffer the pH in a plating bath. It is also used as a complexing agent in depositing metals on substrates. Additives such as ascorbic acid and citric acid improve deposition rate and help in forming finer grain structure, controlling growth, nucleation and alignment of crystal phases during plating [34,35]. Swaminathan et al. [35] investigated the efficiency of an alkaline water electrolysis process in the presence of citric acid. They observed that the appropriate amount of citric acid added achieved better current density than the solution without. This is because citric acid prevents the precipitation of Ni(OH)_2_ from getting adsorbed on the electrode surface to passivate the electrode thereby reducing the kinetics of the reactions. They also attributed the improvement of the material to citric acid wettability and bridging ability in transferring ions between the electrolyte and metal surface. Furthermore, Lertpanyapornchai et al. [36] synthesised mesoporous SrTiO_3_ perovskite via the sol–gel combustion method using citric acid as a metal ion complexing agent. They observed enhanced surface area and larger pore volume. They concluded that the presence of citric acid promoted the formation of high-purity perovskite crystallites than materials without complexing agents. This process of electrodeposition improves the catalytic activity and stability of the electrode owing to the thin films.

Billican et al. [8] electrodeposited BFO on Si/Ti/Cu substrate and carried out characterisation to investigate its ferromagnetic properties. Haromae and Pattananuwat [23] electrodeposited BFO on FTO and probed its properties as a supercapacitor. In this work, we prepared a BFO photoanode by electrodeposition on FTO for the PEC degradation of organic pollutants in water. To the best of our knowledge, the preparation of BFO/FTO by electrodeposition as a photoanode for PEC towards water treatment has not been reported. Furthermore, we characterised this BFO photoanode to investigate its response to light and PEC properties. Physical data from impedance spectroscopy and photocurrent measurements on the electrodeposited films are presented for the first time. We also report the use of citric acid as an alternative complexing agent for BFO electrodeposition. We explore the effect of electrodeposition time (a measure of the film thickness) on the morphology and optical and electrochemical properties of BFO. To demonstrate its impact in water treatment, methylene blue was used as the model pollutant for the evaluation of the photoelectrocatalytic activity of the photoanode.

## 2. Materials and Methods

### 2.1. Materials

All the chemicals used were of analytical grade. Bismuth nitrate pentahydrate (Bi(NO_3_)_3_·5H_2_O) 98% purity, iron nitrate nonahydrate (Fe(NO_3_)_3_·9H_2_O) 98% purity, sodium hydroxide (NaOH) 98% purity, citric acid ≥ 99.5% purity were obtained from Sigma-Aldrich. Deionised water was employed throughout this study.

### 2.2. Preparation of the Photoanodes

In preparation of BFO thin films, an electrodeposition method was employed to electrodeposit films of BFO onto fluorine-doped tin oxide (FTO) glass (50 mm × 13 mm × 3 mm, surface resistivity of ~8 Ω/sq). A chemical bath solution containing 0.1 M bismuth (III) nitrate (Bi(NO_3_)_3_·5H_2_O) (2.425 g), 0.1 M Iron (III) nitrate (Fe(NO_3_)_3_)·9H_2_O (2.019 g) and 0.2 M citric acid (C_6_H_8_O_7_) (2.1014 g) (a complexing agent) were mixed together in 75 mL deionised water while stirring. The solution was adjusted to pH 10 using NaOH solution and the bath was kept at 60 °C. A constant voltage −1.2 V was applied using Autolab PGSTAT204 (Johannesburg, South Africa) potentiostat in a three-electrode setup for time: 300 s, 600 s, 1200 s, 1800 s and 3600 s. Our synthesis method is similar to that of Haromae and Pattananuwat [22], but they used tartaric acid as a complexing agent and a higher voltage of −1.5 V. The photoanodes were annealed at 600 °C for 2 h under synthetic air in a tubular furnace and were allowed to cool to room temperature. The electrodes/photoanodes prepared were labelled according to the electrodeposition time.

### 2.3. Structural and Morphology Characterisation of the Prepared Electrodes

X-ray diffractometer (XRD) from Malvern, United Kingdom was used to study the structural properties and crystallite size of the materials. The morphology and the elemental distribution study of the materials were carried out using field emission scanning electron microscopy (JEOL JSM-7500F, JEOL Ltd., Tokyo, Japan). The film thickness of the photoanodes was measured by viewing the cross-sectional area of the electrodes fractured in liquid nitrogen. The optical properties of the electrodes were studied using the UV–vis diffuse reflectance spectroscopy (DRS) with Shimadzu 2450 spectrophotometer using BaSO_4_ as a reflectance standard. 

### 2.4. Electrochemical and Photoelectrochemical Experiments

An Autolab PGSTAT204 (The Netherlands) potentiostat with three-electrode system and 100 W xenon lamp (solar simulator) were employed for all electrochemical analysis and photoelectrochemical experiments. The electrodes (in a quartz cell) were fixed facing the incident light from the simulator within 10 cm distance. Linear sweep voltammetry was carried out in dark and light between a potential of −0.2 to 1.5 V in a 0.2 M Na_2_SO_4_ solution at scan rate of 0.1 V/s. Chronoamperometry (photocurrent response) experiment was carried out in 0.2 M Na_2_SO_4_ solution. Impedance spectroscopy was evaluated in 5 mM solution of [Fe(CN)_6_]^3−/4−^ salt prepared in a 0.1 M KCl solution with external biased potential of +0.23 V, amplitude of 0.01 V and frequency range of 0.1 Hz and 100 kHz.

### 2.5. Photoelectrocatalytic Degradation of Methylene Blue

The efficiency of the photoanodes was evaluated by measuring the rate of degradation of methylene blue dye (5 mg/L) using photoelectrocatalysis. For each electrode 5 mg/L of methylene blue was prepared in a 0.1 M solution of Na_2_SO_4_ as a supporting electrolyte in a 100 mL quartz glass for 180 min. Electrochemical degradation experiments were carried out in the absence of light but with the application of a bias potential at the photoanode. Photocatalysis was carried out in the presence of light and photoanode. Photolysis was carried out in the presence of light and the absence of the BFO catalyst; while in adsorption (dark) process, the photoanode was suspended in the solution in the absence of light. Other parameters such as stirring speed were kept constant for all the processes.

Aliquots are taken from the reactor at intervals with the aid of a syringe and analysed with UV–vis spectrophotometer to determine the extent of methylene blue removal. Total organic carbon (TOC) removal was evaluated and recorded using TOC analyser (Teledyne Tekmar TOC fusion).

## 3. Results and Discussion

The setup used for the electrodeposition is shown in Figure 1a. All the electrodeposited samples were black in colour at room temperature prior to annealing under synthetic air in a tubular furnace at 600 °C for 2 h. However, after annealing, the colour changed to reddish brown (Figure 1b). 

### 3.1. XRD Analysis

XRD patterns of all electrodeposited photoanodes are shown in Figure 1c. The fluorine-doped tin oxide (FTO) used as the substrate was indexed to the tetragonal phase of SnO_2_ with P42/mnm space group (ICDD no: 04-003-5853). The diffraction peaks were indexed to the rhombohedral perovskite phase of BFO with the R3c space group. The 300 s photoanode shows major peaks at 31.7° and 32.0° which was assigned to (104) and (110) crystal plane of rhombohedral BiFeO_3_ (ICDD PDF 04-014-7269). The XRD patterns showed some secondary phases which were identified to be orthorhombic phase of Bi_2_Fe_4_O_9_ with space group Pbam (ICDD PDF 00-025-0090) that formed along with BFO as the electrodeposition times increases as observed in some reports [8,37]. For example, Bilican et al. [8] synthesised BFO using the electrodeposition method (annealed for 500, 600, 700 and 800 °C) and observed similar peaks which they attributed to the secondary phase. Figure 1d shows the XRD pattern of 300 s photoanode with the BFO prominent peaks at 2θ = 31.7° and 32.0° as seen in the inset which was enlarged between 2θ = 25° to 35°. The peak at 45.7° in the main graph is assigned to BFO. Photoanode electrodeposited at 300 s which is the optimal electrode, shows the least prominent secondary phase as compared to 3600 s (Figure 1c). While we cannot ascertain the effect of this secondary phase in the PEC response, the fact that the 300 s photoanode showed the best visible light responses and PEC degradation performance (to be discussed later in Section 3.5) suggests that the PEC performance is from the BFO and not the secondary phases. The crystallite size (D) in the (hkl) plane direction was estimated from the full width at half maximum (FWHM) of the XRD at peak position 32° using the Debye Scherrer formula (Equation (1)).
(1)$$D=\frac{0.9\lambda }{\beta cos\theta }$$
where λ is the X-ray wavelength (λ = 1.5406 Å), β is the estimated angular width at half maximum (FHWM) of the peak and θ is the Bragg angle [38]. 

At plane (104), the crystallite size increased from 21.6 nm to 25.5 nm with increasing deposition time from 300 s to 1800 s (Table 1). However, at 3600 s, the crystallite size dropped to 24.7 nm. The slight change in the crystallite size could be due to compaction and multi-layers of BFO films over time [39]. We also observed that the peak intensities of 1800 s and 3600 s electrodes were more pronounced than electrodes deposited at 300 s, 600 s, and 1200 s because of the larger crystallite size.

### 3.2. FESEM Analysis

The impact of the electrodeposition time on the morphology of the epitaxial thin film of BFO is seen by SEM micrographs in Figure 2a(i–v). The SEM images display different morphologies for each photoanode: Figure 2a(i) photoanode shows irregular shapes and Figure 2a(ii), shows stick and flowery-like shapes. Photoanode at Figure 2a(iii) displays a mixture of cube and round-like shapes. A stick-like shape dominated, and flowery shapes were observed in Figure 2a(iv) and Figure 2a(v), respectively. While the reason for the different morphologies at changing deposition times is not clear, it is possible that each layer of deposition acts as a nucleating or seeding surface to another layer and the combination alters the morphology. As seen with XRD patterns, the peaks also differ with electrodeposition time. We observed secondary phases as electrodeposition time increased. The change in morphology and structure with deposition time however provides a possible reason for the differences in photoelectrocatalytic properties. Morphology changes in nanostructured materials with resulting changes in photocatalytic or photoelectrocatalytic responses have been reported [25,40,41]. 

The film thickness of the electrodeposited BFO was calculated by measuring the mean of the cross-sectional area of the electrodes at different positions (Figure 2b(i–v)). The mean film thickness of 300 s, 600 s, 1200 s, 1800 s and 3600 s electrodes were 2.54 μm, 5.15 μm, 6.63 μm, 10.29 μm and 14.14 μm (see Figure 2b inset). As expected, the film thickness increased with increasing electrodeposition time, and it is in agreement with the results obtained by measuring the mass of the FTO before and after electrodeposition in Table 2. Figure 2c,d show the line scan of the EDX and the EDX image of the elemental composition, respectively. The micrograph inserted in Figure 2c shows the actual contrast of where the measurement was taken. It displays the course of Bi Mα1, Fe Kα1 and O-Kα X-rays across the specimen. It is interesting to note that the presence of the Bi Mα1 signal is strongly enhanced followed by O Kα1 and Fe Kα1. As seen in Figure 2d, the EDX image successfully shows the elemental composition of BFO on the 1200 s photoanode. The distance within the distribution in the scan line graph (~11 μm) corresponds with the film thickness of 1800 s photoanode (10.29 μm).

### 3.3. Optical Properties

The UV–vis absorption spectra of samples prepared at different electrodeposition times are presented in Figure 3a. All samples show absorption in the visible light region. However, samples electrodeposited within a shorter time show higher absorbance between 450 and 600 nm while samples electrodeposited at longer times absorb more on the ultraviolet region (280 and 380 nm). This shows that thin film enhances the visible light absorption capacity of BiFeO_3_. This trend is also noticed in the photoelectrochemical analysis to be discussed in later sections. 

The band gaps of all electrodeposited samples were calculated using Tauc’s equation (Equation (2)) [42]:(2)$$\alpha hv={A(hv-E_{g})}^{n}$$
where v is the frequency of the incident photon, h is Planck’s constant, A is a proportionality constant, and n = 1/2 for material with direct band gaps and n = 2 for material with indirect band gaps. Bismuth ferrite is a material with a direct band gap [17,43], hence n = 1/2. Figure 3b indicates the difference in band gap energy at different electrodeposition times as extrapolated from a plot of (αhv)^2^ against hv on x axis = 0. We observed an increase in energy band gaps from 1.73 eV to 2.06 eV as electrodeposition time increases from 300 s to 3600 s. We attribute this shift in band gap to the easy penetration of light into the thin films than multi-layer films and the change in morphology. This observation may be more visible with a reduction of active sites in the photocatalytic activity of photoanodes electrodeposited at a longer time. These findings are supported by the film thickness results measured with FESEM. Some reports on electrodeposition methods have shown that band gaps decrease with an increase in electrodeposition time [33,44,45], which they attributed to grain size in the polycrystalline film. However, in our study, we observed an increase in band gap with an increase in film thickness as time increased. Agglomeration in most cases can lead to an increase in defects and disorder in the crystallinity, which increases the density of the localised states in the band structure thereby increasing the band gaps. Momeni and co-workers [46] recorded a slight increase in band gaps of BiVO_4_ as electrodeposition time increased. Similarly, Maleki et al. [36] studied the effect of the thickness of BFO thin films; they observed that the band gap increased as the thickness of the film increased. These results correlate with what we observed in this study.

### 3.4. Electrochemical and Photoelectrochemical Studies

Photocurrent density was obtained using linear sweep voltammetry (LSV) and chronoamperometry experiments in the presence and absence of irradiated visible light. The photocurrent density of all the photoanodes using LSV when toggled between dark (off) and light (on) is shown in Figure 4a, and the photocurrent density is inversely proportional to deposition time. To further understand the photocurrent response of all photoanodes, chronoamperometry experiments were carried out (Figure 4b). The in-text shows the values of the current density of all photoanodes. All photoanodes responded to the solar light as confirmed by the increase in photocurrent upon the stimulation of light. The decrease in photocurrent density of electrodes electrodeposited at a longer time as observed by the linear sweep and chronoamperometry analysis could be the thickness of the film over time which could cause insufficient penetration of visible light. This agrees with the results observed in the optical analysis. Notably, controlling the BFO film thickness as observed by the literature could have a profound effect on the photocurrent density, subsequently leading to higher efficiency in the photoelectrocatalytic activity process. The increase in the photocurrent response is an indication that thin film photoanodes could enhance interfacial charge transfer and consequently inhibit the recombination of electron–hole pairs [47]. This observation of an increase in current density in thin films with shorter electrodeposition time is in good agreement with the results reported by Yam et al. [48] and Shafi et al. [49] who attributed their observations to an increase in surface area and surface defects. 

To gain more insight into charge carrier properties and PEC performance of the BFO thin film, electrochemical impedance spectroscopy (EIS) was used. The Nyquist plots for the prepared photoanodes are shown in Figure 4c; the impedance data were fitted with the Randle circuit (Figure 4c (inset)). Rs, Rct, Zw and Cdl represent the solution resistance, the charge transfer resistance, the Warburg diffusion and the double-layer capacitance, respectively. All the photoanodes exhibit low charge transfer resistance, however, the charge transfer resistance increases as the electrodeposition time increases with 300 s, 600 s, 1200 s, 1800 s and 3600 s exhibiting 350, 425, 390, 690 and 920 Ω, respectively. The photoanode with rapid charge transfer and excellent separation of electron–hole pairs are evident by smaller semicircle or low charge transfer resistance [50]. The photoanodes electrodeposited at lower times such as 300 s, 600 s, and 1200 s showed lower charge transfer resistance and this result is ascribed to greater current mobility and diffusion, and a good demonstration of effective charge transfer and separation of charge carriers as compared to others [51].

### 3.5. Photoelectrocatalytic Degradation Experiments

To study the effect of the electrodeposition time of BFO films on photoelectrocatalysis, the photoanodes: 300 s, 600 s, 1200 s, 1800 s and 3600 s were used for the photoelectrocatalytic degradation of 5 mg/L methylene blue (MB). The change in absorption at λ_max_ = 663 nm was used to measure the extent of degradation of MB. The degradation efficiencies of 97.9%, 89.7%, 78.13%, 80% and 70% for 300 s, 600 s, 1200 s, 1800 s and 3600 s, respectively, were calculated after 180 min (Figure 5a). The high degradation percentages could be attributed to excellent interfacial charge separation and suppression of photogenerated electrons and holes as recorded in the EIS experiment. Although, we still observe the trend of a decrease in performance with an increase in electrodeposition time. Possible explanations for this observation are as follows: (i) increase in e^−^/h^+^ pair production due to greater harvest (or absorption) of visible light through the thin film as seen in Figure 3a for 300 s deposition. (ii) Higher current mobility which led to rapid electron transfer as seen in Figure 4c where charge transfer resistance decreases with a decrease in electrodeposition time. (iii) Increase in thickness could lead to agglomeration which hinders visible light penetration thereby reducing PEC degradation efficiency. To obtain the degradation rate constants, the results were fitted into a pseudo-first-order kinetic model as shown in Figure 5b. The apparent reaction rate constant for 300 s, 600 s, 1200 s, 1800 s and 3600 s photoanodes are 18.0 × 10^−3^ min^−1^ (R^2^ = 0.9460), 12.5 × 10^−3^ min^−1^ (R^2^ = 0.9476), 8.5 × 10^−3^ min^−1^ (R^2^ = 0.9400), 8.9 × 10^−3^ min^−1^ (R^2^ = 0.9820), 6.1 × 10^−3^ min^−1^ (R^2^ = 0.9457). These results reveal that the reaction rate was fastest with photoanodes electrodeposited at a shorter time and the coefficient of regression of all photoanodes was well-fitted into the first-order kinetic model. Thin films are known to possess more active sites which leads to improvement in charge carrier separation thereby reducing the tendency of recombination of photogenerated electrons and holes [52]. The contributions of adsorption, light, voltage and catalyst to the PEC degradation process were determined with some control experiments such as adsorption (dark), photolysis, photocatalysis and electrocatalysis using the optimal electrode, i.e., at 300 s (Figure 5c). The removal of methylene blue was conducted to investigate the adsorption effect of the catalyst and the synergistic effect of light and the applied bias potential in the performance of photoelectrocatalysis. The removal rate of 5 mg/L of methylene blue in adsorption, photolysis, photocatalysis, electrocatalysis and photoelectrocatalysis processes were calculated to be 6.5%, 13.0%, 32.3%, 47.1% and 97.9%. The observed removal rate of adsorption and photolysis processes were low due to the immobilisation of the catalyst onto a substrate as opposed to a powdery catalyst thereby reducing the active sites due to low surface area. To obtain the degradation rate constants, the results were fitted into the pseudo-first-order kinetic model (Figure 5d). The apparent reaction rate constant for adsorption, photolysis, photocatalysis, electrocatalysis and photoelectrocatalysis were 3.2 × 10^−4^ min^−1^ (R^2^ = 0.9998), 13.6 × 10^−4^ min^−1^ (R^2^ = 0.9523), 21.4 × 10^−4^ min^−1^ (R^2^ = 0.9230), 36.0 × 10^−4^ min^−1^ (R^2^ = 0.9648) and 19.0 × 10^−3^ min^−1^ (R^2^ = 0.9381). These suggest that the rate of reaction was fastest at the photoelectrocatalysis process. 

To further understand the advantage of PEC over photocatalysis (PC) and electrocatalysis (EC), the degree of electrochemical enhancement (E) and the degree of process synergy (S) were calculated using Equations (3) and (4).
(3)$$E=\frac{{(K}_{PEC}-K_{PC})}{K_{PEC}}$$
(4)$$S=\frac{\left(K_{PEC}-(K_{PC}+K_{EC})\right)}{K_{PEC}}$$

The degree of enhancement (E), calculated to be 0.8874 (88.7%), reveals that the use of bias potential enhances the performance as can be seen from the photoelectrocatalytic degradation process of MB. The degree of process synergy obtained was 0.6980 (higher than zero) confirming that the improved performance observed in PEC degradation is more than the summation of PC and EC processes. In other words, the result proves the synergistic benefit of both processes [17,53]. The extent of mineralisation of the methylene blue was monitored by total organic carbon (TOC) after 180 min. TOC removal for 300 s, 600 s, 1200 s, 1800 s and 3600 s photoanodes were 63%, 52%, 44%, 46% and 35%, respectively (Figure 5e). Again, these results show the better performance of thinner BFO film at 300 s over a longer deposition time as observed in the degradation efficiency measured by UV–vis spectrophotometry. The reason for this TOC trend is the same as that put forward earlier in the section for degradation percentage because the TOC is based on the same photoelectrochemical oxidation principle. 

Stability and reusability are pivotal indices for selecting an efficient photoanode in PEC. After six cycles of degradation at 2 V bias potential, the change in degradation was less than 2% suggesting good photoanode stability (Figure 5f). To further probe into the integrity of the BFO photoanode after six cycles, we examined the XRD of before and after degradation of MB using the optimal electrode (Figure 5g). The major diffraction peaks at 31.7 and 32.0° indexed to (104) and (110) crystal planes of BFO are still maintained in the photoanode after six runs. This is an indication of stability and the retention of the structural integrity of the prepared catalyst. 

## 4. Conclusions

In summary, visible light active BiFeO_3_ photoanodes were successfully electrodeposited onto FTO using citric acid as a complexing agent for 300 s, 600 s, 1200 s, 1800 s and 3600 s. The variation of electrodeposition greatly impacted the photoelectrochemical properties of BFO. The bandgap of the photoanodes increases as electrodeposition time and film thickness increase. In addition, with the increased surface area associated with thin films, higher photocurrent responses and low charge transfer resistance were observed with photoanodes electrodeposited at a shorter time. The photoelectrocatalytic removal rate of 97.9%, 89.7%, 78.13%, 80% and 70% for 300 s, 600 s, 1200 s, 1800 s and 3600 s, respectively, were observed for PEC degradation of MB. The PEC system gave a better degradation percentage than the adsorption, photolysis, photocatalysis and electrocatalysis processes. The electrochemical enhancement process and degree of process synergy proved the pivotal role the combination of light and bias potential can play in the PEC system. For better catalyst stability, this study has shown that perovskite photoanode can be prepared by the electrodeposition process using citric acid as a complexing agent for water treatment via photoelectrocatalysis. This electrodeposition method served to enhance electrode stability and improvement in performance. The elevated performance of 300 s photoanode is attributed to better light harvesting, and charge carrier diffusion from the increased surface area. We have demonstrated that in the electrodeposition method, electrodeposition time, the thickness of the electrodeposited film and morphology have an effect on the photoelectrochemical properties of BFO films formed on the FTO substrate.

## Figures and Tables

**Figure 1 materials-16-02769-f001:**
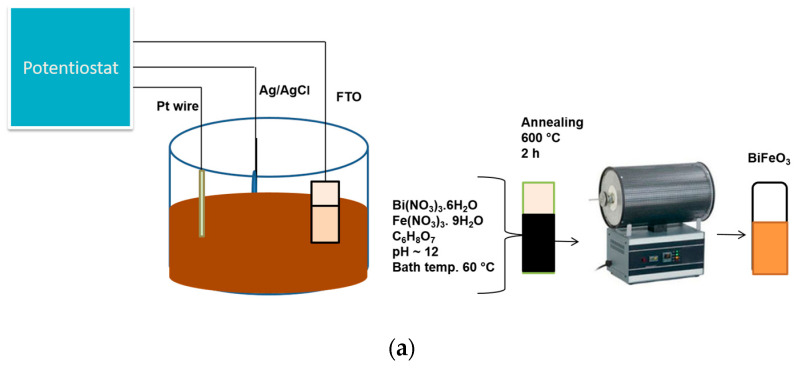
(**a**) Schematic diagram for the preparation of BFO film using simple chemical bath electrodeposition method, (**b**) appearance of BFO films before and after annealing (**c**) XRD patterns of FTO and FTO/BFO at 300 s, 600 s, 1200 s, 1800 s, 3600 s, (**d**) XRD pattern of 300 s and enlarged image (inset) of 300 s photoanode.

**Figure 2 materials-16-02769-f002:**
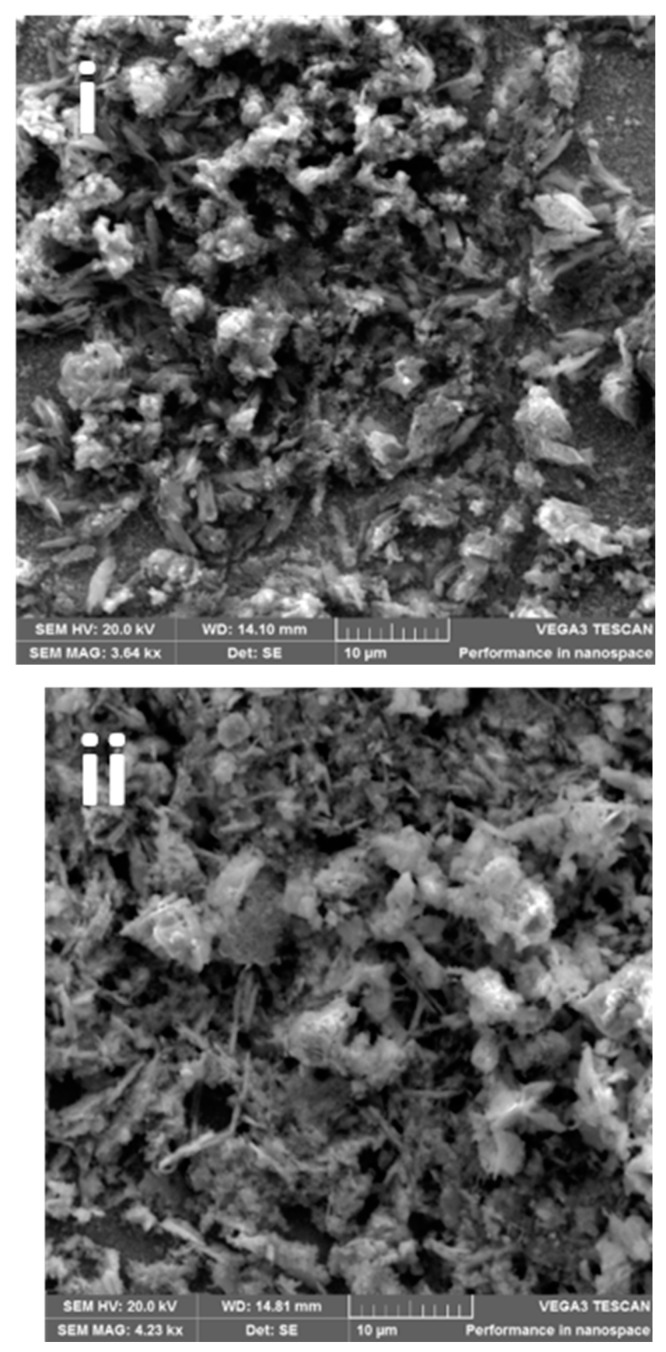
(**a**(**i**–**v**)) SEM images, (**b**(**i**–**v**)) FESEM film thickness of (**b**(**i**)) = 300 s, (**b**(**ii**)) = 600 s, (**b**(**iii**)) = 1200 s, (**b**(**iv**)) = 1800 s and (**b**(**v**)) = 3600 photoanodes, (**c**) EDX line scan (20 KV beam voltage) of a cross-sectional area of 1200 s BFO photoanode, (**d**) EDX image showing the elemental composition.

**Figure 3 materials-16-02769-f003:**
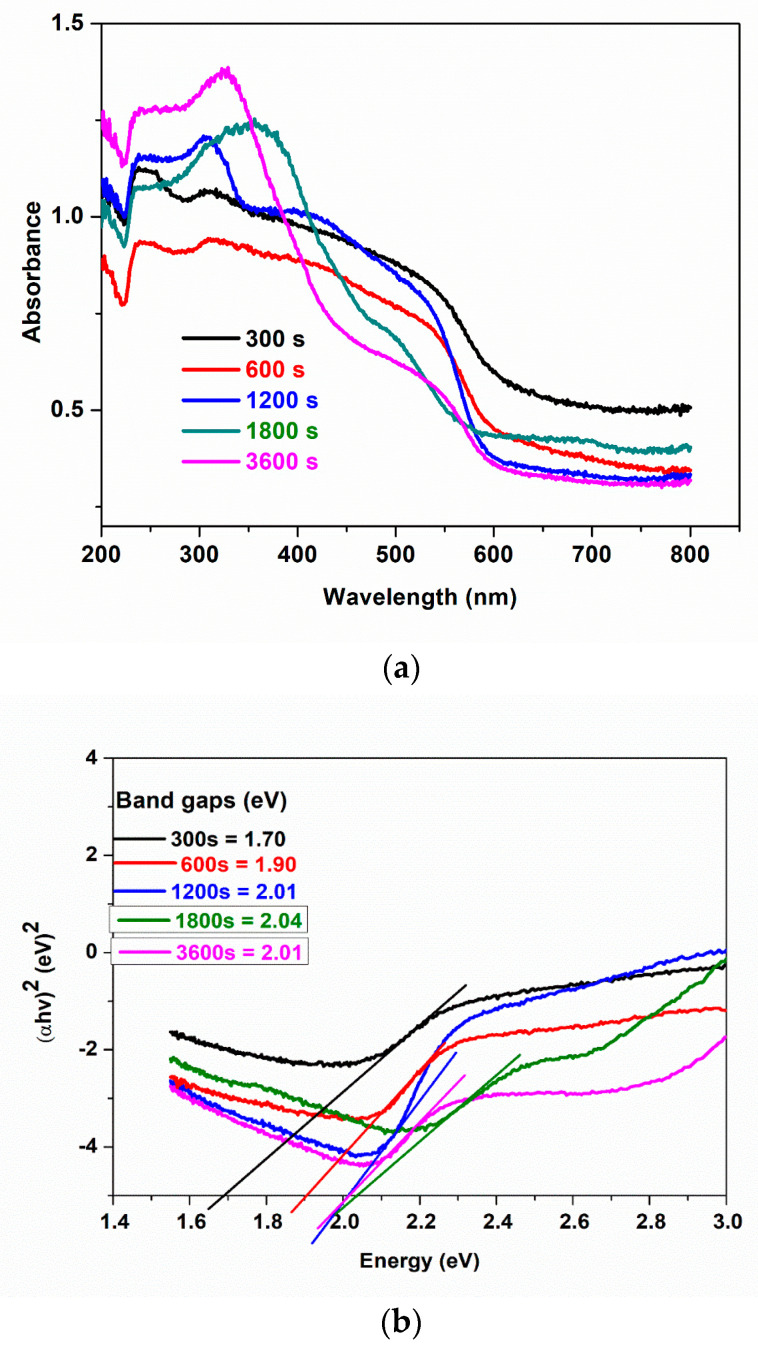
(**a**) UV−visible diffuse reflectance spectra, (**b**) the band gap energies of 300 s, 600 s, 1200 s, 1800 s, 3600 s.

**Figure 4 materials-16-02769-f004:**
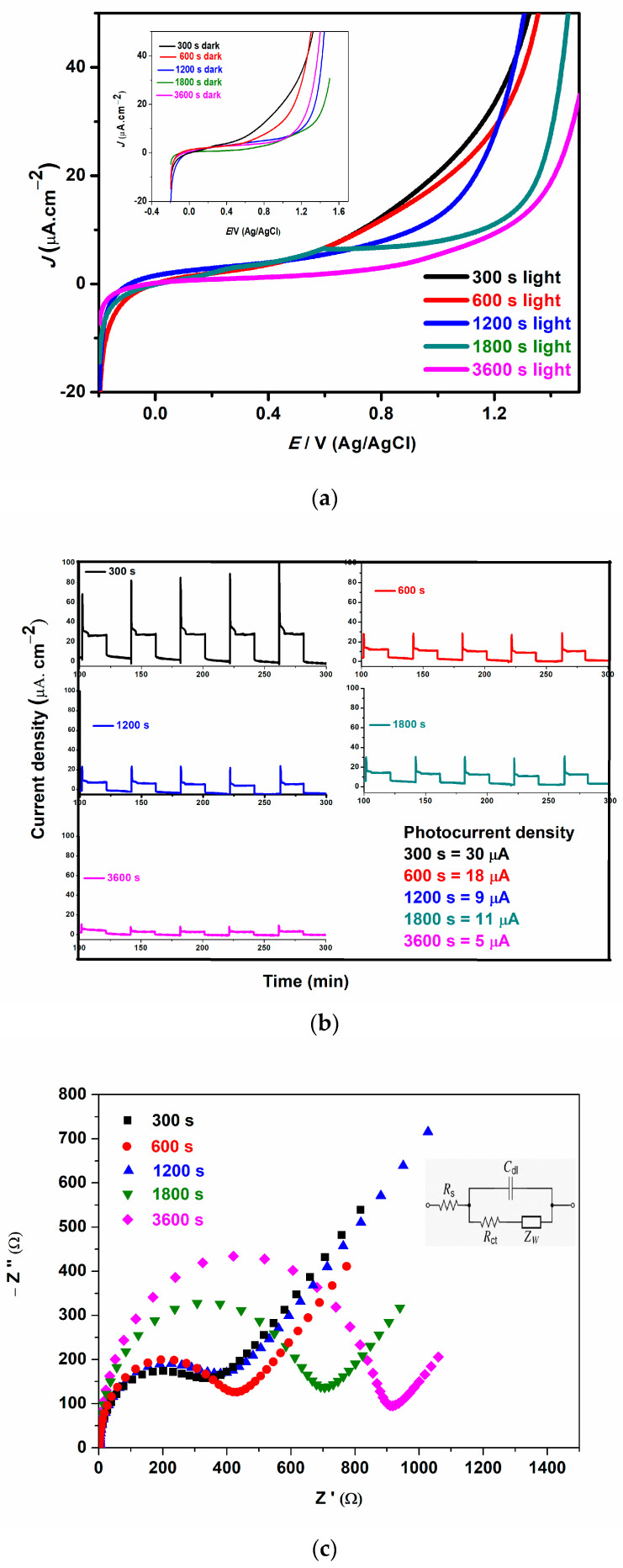
(**a**) Linear sweep voltammogram (LSV) under light with a scan rate of 0.1 V/s (inset: LSV plot under dark, (**b**) photocurrent response, (**c**) electrochemical impedance spectroscopy for 300 s, 600 s, 1200 s, 1800 s, 3600 s photoanodes.

**Figure 5 materials-16-02769-f005:**
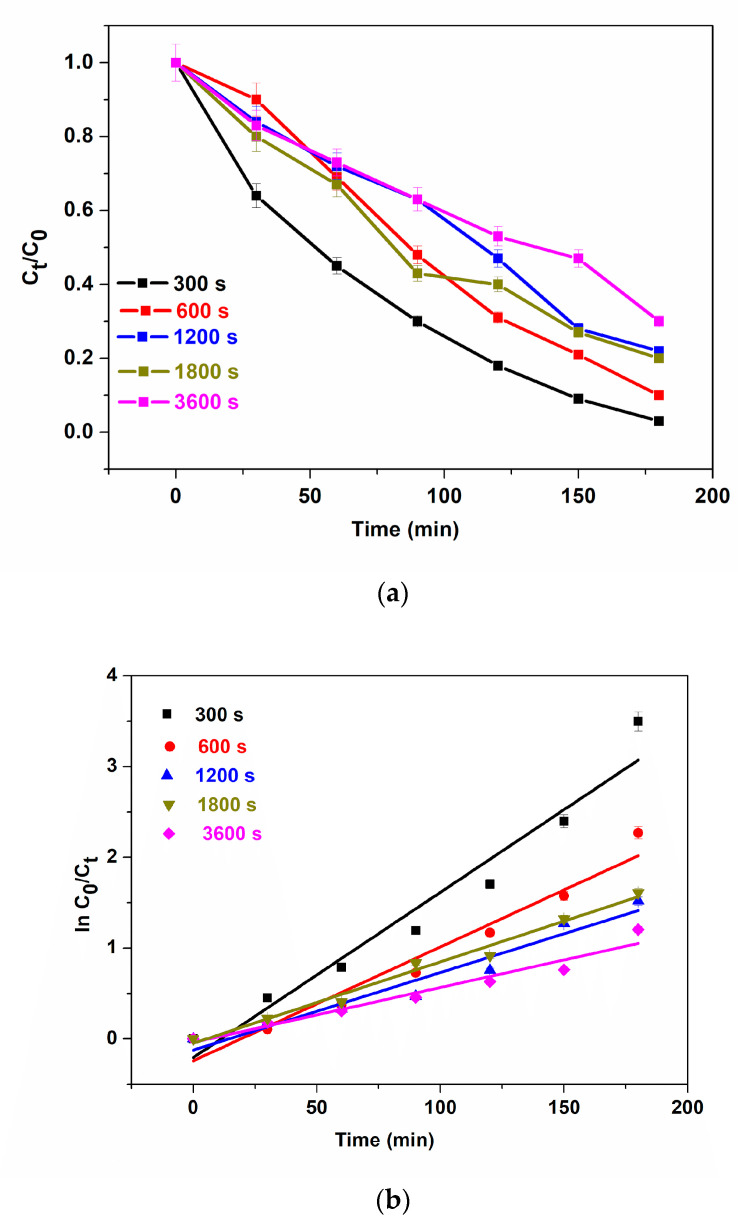
(**a**) Photoelectrocatalytic degradation (**b**) kinetic plots for degradation of MB using 300 s, 600 s, 1200 s, 1800 s and 3600 s, (**c**) different processes of degradation, (**d**) kinetic plot for different processes of degradation of MB using 300 s photoanode, (**e**) TOC removal of methylene blue, (**f**) reusability study, (**g**) XRD patterns of BFO before and after 6 cycle degradation (5 ppm, 2 V, methylene blue, 3 h).

**Table 1 materials-16-02769-t001:** XRD Measurements (from Figure 1c).

t_d_ (s)	Bragg Angle 2θ (°)	FHWM	Crystalline Size (nm)
300	31.9	3.5319	21.62
600	32.15	3.4453	22.15
1200	32.20	3.1080	24.55
1800	31.90	2.9847	25.59
3600	32.20	3.0835	24.75

**Table 2 materials-16-02769-t002:** Difference in mass of BFO photoanodes at different electrodeposition times.

Deposition Time (t_d_ (s))	300	600	1200	1800	3600
Before deposition (g)	3.5600	4.5902	3.3141	3.4849	4.8843
After deposition (g)	3.5688	4.6069	3.3350	3.5394	4.9539
Difference (g)	0.0088	0.0167	0.0209	0.0545	0.0696

## Data Availability

The data presented in this study are available on request from the corresponding author.

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
