# Peer review of "Photoelectrocatalytic Degradation of Methylene Blue on Electrodeposited Bismuth Ferrite Perovskite Films"

_materials, 2023, doi:10.3390/ma16072769_

Round 1

Reviewer 1 Report

The authors presented an electrodeposited BiFeO3 thin film on a FTO substrate for photoelectrocatalytic degradation of methylene blue. It is an interesting study and fits well in this special issue after revisions.

The XRD method should be extended including scanning range and resolution. Also the method used to determine crystallite size should be added here.

For the XRD results, I suggest to thoroughly analyze the XRD pattern as it seems several impurities not related to the perovskite phase exist. If this is the case, how will the authors determine the effect of the perovskite phase on properties and not the effect of other phases?

The SEM and FESEM images includen EDX mapping need great improvement to be easier read. Currently the text and atomic symbols are difficult to read.

It should be thorougly discussed why an increased deposition time results in less degradation of methylene blue. In addition to this, what is the criteria to obtain an excellent activity? Is 70 % degradation excellent?

Did the authors investigate potential leaching from the perovskites during experiment to further consider the reusability and potential for industrial use?

Some minor corrections:

Please unify the Author [REF] et al or Author et al ... text ... [REF] through the manuscript.

The FTO glass abbreviation should be mentioned. What is the FTO glass?

When stating "A procedure reported [22] with modification was used." in line 117, it should be described how this method differs from the former OR just describe the method used here without citing the former procedure.

The UV-vis wavelength used to determine methylene blue removal should be added.

Figure caption 5 states c) twice. The last one should be corrected to (e).

Reviewer 2 Report

I reviewed your manuscript “Bismuth ferrite perovskite photoanode prepared by electro deposition in citric acid: characterisation and photoelectrocatalytic application in the degradation of methylene blue dye in water.” very judiciously. The work carried out in the manuscript is very interesting and seems scientifically logical. The authors have added good technical value and knowledge to remove the water waste specially dyes from the pure water recourses. However, there are numerous inaccuracies in this work and before publication, it needs to restructure the research manuscript properly and diligently as the current presentation is not acceptable. Therefore, I would like to recommend this article for "Major Revision".

1.    Title

The title is too long. It should be short, meaningful and attractive. Please modify the title.

2.    Introduction

The novelty of this work was not specified; authors should discuss the novelty of their work in the introduction section. The author should made comparison between their photocatalysts and already reported photocatalysts like hexaferrites materials. The author should also read the research articles related to the wastewater purification, chemical physics letters 805, (2022), 139939, 431–440, New J. Chem., 2022,46, 19848 and doi.org/10.1080/03067319.2022.2032014

3.    Materials and methods

The percentage purity of all the chemicals utilized must be reported. Moreover, photodegradation experiment should also in the material and method section. No need to give the detail of instrumentation. Just write a paragraph that describes all your characterization techniques.

4.    XRD analysis

Section 3 should be heading as “Results and Discussion”. “It has been reported that citric acid en-168 courages finer grain and pure perovskite phase formation” Justify this comment with proper explanation. What is formation22 in line 169? All the BFO carried out structural transformation after the 2 theta value of 45. The author must discuss this phenomena. Appearance of several new peaks after 600s have questioned mark on the purity of the samples, because the photodegradation of dyes highly depend upon the purity of the samples. Moreover, quality of the XRD pic is very poor. I recommend to revise the XRD analysis carefully.

5.     SEM& EDX

SEM images are not clear and aligned scientifically. There is no connection between SEM and XRD analysis. EDX portion must include the atomic as well as weight %. Please use the scientific and recommended font style and size in all the figures. Write all the equation in scientific way.

6.     Photodegradation

The line intercept are wrongly drawn and not exist in the frame along x axis.  I strongly recommend to provide XRD analysis of the samples after photodegradation experiments because photocatalytic properties of the synthesized samples are highly sensitive to the purity of the samples.

7.    Comparison should be made between already reported and author’s submitted article.

8.    The author do not explain properly the increase in % photodegradation for 300s to 3600s samples. Please explain properly otherwise I will not able to recommend the manuscript.

9.     I suggest the author to add the digital Photographs of all adsorption and degradation experiments (before degradation and after degradation photograph).

10. Lot of literature available where the degradation and adsorption of dyes has been reported in similar duration, therefore the novelty ion this work is missing. The author should explain the novelty in proper way.

11.  Read the manuscript diligently and remove all the typographical (like R2 should be R2),  font style and font size mistakes

12. Revise the conclusion section and add meaningful and numerical values to make it more attractive and easy understanding for new researchers.

Round 2

Reviewer 1 Report

The authors did a thorough revision of the manuscript. I have one last comment. 

As secondary phases occur based on the XRD pattern, is the photoelectrochemical effect caused by the BFO perovskite, or is some of the secondary phases such as Bi2Fe4O9 the key? Please address this in the manuscript.

Author Response

Comments and Suggestions for Authors

The authors did a thorough revision of the manuscript. I have one last comment. 

Response: Thank you.

As secondary phases occur based on the XRD pattern, is the photoelectrochemical effect caused by the BFO perovskite, or is some of the secondary phases such as Bi2Fe4O9 the key? Please address this in the manuscript.

Response: This is a good comment. We cannot for not fully ascertain the role of the secondary phase in the PEC degradation. We believe that the PEC is due to the BFO because the 300 s electrode has the least amount of secondary phase but gave the highest degradation. If the secondary phases were responsible, one would have observed the best degradation from the 3600 s electrode but it was not so in our case. Please see our revised discussion on XRD.

Some reports on the photocatalytic application of BFO also reported the presence of some secondary phase [ref 8,37].

Reviewer 2 Report

Quality of Figure 2(a&b) should be improved before publication

References format should be corrected

Author Response

Quality of Figure 2(a&b) should be improved before publication.

Response: Thank you. We have improved the image. For 2b, the main reference is the thickness which we have written out on the image for visibility.

References format should be corrected

Response: We have corrected it.